# Regenerative Therapy for Corneal Scarring Disorders

**DOI:** 10.3390/biomedicines12030649

**Published:** 2024-03-14

**Authors:** Christine Chandran, Mithun Santra, Elizabeth Rubin, Moira L. Geary, Gary Hin-Fai Yam

**Affiliations:** 1Corneal Regeneration Laboratory, Department of Ophthalmology, Mercy Vision Institute, University of Pittsburgh School of Medicine, Pittsburgh, PA 15219, USA; chc337@pitt.edu (C.C.); mithun.santra@pitt.edu (M.S.); eer35@pitt.edu (E.R.); mlo39@pitt.edu (M.L.G.); 2McGowan Institute for Regenerative Medicine, University of Pittsburgh, Pittsburgh, PA 15219, USA

**Keywords:** cornea scarring, regenerative therapy, corneal stromal cells, extracellular matrix, gene therapy, tissue engineering

## Abstract

The cornea is a transparent and vitally multifaceted component of the eye, playing a pivotal role in vision and ocular health. It has primary refractive and protective functions. Typical corneal dysfunctions include opacities and deformities that result from injuries, infections, or other medical conditions. These can significantly impair vision. The conventional challenges in managing corneal ailments include the limited regenerative capacity (except corneal epithelium), immune response after donor tissue transplantation, a risk of long-term graft rejection, and the global shortage of transplantable donor materials. This review delves into the intricate composition of the cornea, the landscape of corneal regeneration, and the multifaceted repercussions of scar-related pathologies. It will elucidate the etiology and types of dysfunctions, assess current treatments and their limitations, and explore the potential of regenerative therapy that has emerged in both in vivo and clinical trials. This review will shed light on existing gaps in corneal disorder management and discuss the feasibility and challenges of advancing regenerative therapies for corneal stromal scarring.

## 1. Introduction

Various insults to the cornea, including physical injuries, chemical burns, inflammation, and diseases, cause tissue damage leading to a complex cascade of tissue inflammatory and fibrotic responses, as well as tissue remodeling and healing to prevent further damage and restore tissue integrity and functions. Tissue remodeling mainly involves the production of a new extracellular matrix (ECM) to close the wound region and protect the underlying tissue from infection and further damage [1]. The excessive deposition of ECM components and matrix remodeling result in corneal opacities, eventually leading to a scarred cornea, blocking light penetration, and causing visual impairment. Superficial and mild scarring can heal within months and vision can be restored; however, deep, or severe scars usually worsen and lead to corneal blindness.

Corneal scarring is the fourth leading cause of blindness globally, affecting about 2 million people (5.1% of the overall number of blind people). Additionally, in each year, more than 350,000 children are born with or develop infections at a younger age that lead to corneal blindness [2,3]. The social and economic burden on the individual and the wider community is huge, as it tends to affect younger people, unlike other blindness due to cataract and glaucoma. It also disproportionately affects low-income and rural communities due to a high prevalence of communicable diseases (e.g., trachoma and onchocerciasis), a higher risk of injuries from contaminated objects, and limited access to treatments [4].

The treatment to effectively heal a scarred cornea poses significant challenges. Corneal repair to restore corneal transparency and visual recovery encompasses different options, including pharmacological treatments, steroids, immune modulation, and grafting with donor corneal materials to replace the damaged tissue. Current pharmacological (non-surgical) therapies offer a reasonable improvement of corneal haze and mild-to-moderate opacities. Surgical treatments with donor cornea transplantation (penetrating, anterior and deep lamellar keratoplasties) are the current standard of care for severe and permanent opacities and corneal blindness. However, these treatments and the wide use of grafting have been restricted by many factors and challenges. Hence, more research on understanding of the etio- and cyto-pathological mechanisms and pathways promoting corneal stromal wound repair and tissue regeneration that allow a restoration of corneal transparency, functions, and integrity is demanded. This review highlights the recent advances in the regenerative approaches to treating corneal scarring.

## 2. Cornea—Structure and Functions

The ocular surface consists of the cornea, conjunctiva, lacrimal glands, and eyelids. The cornea is a dome-shaped, transparent, and avascular tissue that covers the front part of the eye (Figure 1A). A healthy cornea transmits light and acts as a refractive medium to focus light rays through the lens to the retina, providing two-thirds of the refractive power to the eye. Additionally, the cornea is mechanically strong, serving as a barrier to protect the inner eye tissues [5]. All of these features are imperative for normal vision.

Composed of distinct cellular and acellular layers with unique structural and organizational characteristics, the cornea plays a critical role in the overall visual sensation and acuity. Anatomically, the cornea contains three cellular layers, which are the outermost corneal epithelium (CEpi), middle corneal stroma (CSt), and the innermost corneal endothelium (CEndo) in a sandwich with two acellular layers: the Bowman’s layer and the Descemet’s membrane (Figure 1B). The CEpi acts as a barrier with intercellular tight junctions to protect the corneal tissue against bacteria, microbes, chemicals, and maintain a smooth ocular surface to retain moisture and the tear film. It is highly regenerative, due to the limbal stem cells present at the peripheral limbus. This allows for CEpi regeneration in daily turnover (homeostasis) and for rapid ad integrum healing in response to injury (regeneration).

The corneal stroma (CSt) constitutes 80–90% of the entire corneal volume and primarily consists of parallelly aligned stromal lamellae composed of collagen I, IV, V, and proteoglycans (keratocan, lumican, mimecan, and decorin. The orthogonal arrangement of stromal lamellae enables light passage with minimal interference and scattering of light and supports the cornea’s mechanical strength and integrity. The corneal stromal keratocytes (CSKs) are the primary stromal cell type located between the stromal lamellae (Figure 1C). These neural crest-derived cells are typically quiescent (very limited proliferation and regeneration) and present dendritic processes connecting the neighboring CSK and forming a highly organized syncytium of cell network. They are essential for the development and maintenance of stromal lamellae by regulating and depositing collagen fibrils and matrix proteins [6]. At the stromal periphery (anterior limbal stroma), the existence of corneal stromal stem cells (CSSCs) represents an adult progenitor population for CSK. Although there is a lack of in situ evidence showing their differentiation to produce CSK inside CSt, different pre-clinical studies have indicated their therapeutic potency to remodel CSt after injury, resulting in stromal regeneration and a recovery of regular lamellar pattern [7,8,9].

The CEndo contains a monolayer of cuboidal CEndo cells, which possess an “active pump–passive leak” function to regulate water and osmolyte content inside CSt [10]. This activity maintains the stromal hydration status and prevents edema, which disrupts the regular arrangement of collagen fibrils and causes light scattering.

## 3. Corneal Stromal Homeostasis and Wound Healing

Corneal stromal homeostasis is a meticulously regulated process that is crucial for maintaining the cornea’s transparency, refractive properties, and mechanical strength. As the primary component of the cornea, the stroma comprises an intricate matrix of collagen fibers, proteoglycans, and other constituents of the extracellular matrix (ECM) [11]. The absence of vasculatures and limited immune surveillance is crucial for maintaining stromal homeostasis and preserving visual acuity.

The corneal stroma (CSt) has the primary site to regulate corneal transparency, which emanates from the organized disposition of collagen fibers and stromal lamellae. The precise arrangement minimizes light scattering, ensuring undisturbed light passage. Any disruption to this stromal architecture, such as traumatic injury or pathological conditions, has the potential to compromise the corneal transparency. Mechanical resilience is a critical aspect of corneal homeostasis, achieved through the collaboration between collagen fibers and proteoglycans furnishing the cornea with tensile strength and resistance to deformation. This mechanical stability is imperative for maintaining corneal structural integrity and withstanding external forces. In addition to collagen, elastic tissue plays a crucial biomechanical role in stromal tissue dynamics. These elastic fibers, composed mainly of fibrillin-1 protein, form bundles of microfibrils. They appear as elastin-containing sheets at the limbus, extending into the stroma and forming fibrillin-rich microfibril bundles. These microfibril bundles, running parallel to the cornea’s surface, are concentrated in the posterior stroma near Descemet’s membrane.

CSKs, as the resident cell type, are fundamental to the synthesis, deposition, and turnover of ECM components. The cells are critical for the homeostasis of a viable CSt through their regulation of collagens and proteoglycans in the stromal matrix. Human adult CSKs are normally quiescent (arrested at G0/G1 stage) with limited proliferation or apoptosis, and there has been an estimated 0.45% cell loss per year [12]. Summarized from different studies, human CSKs express unique molecular markers, including keratocan, lumican, and aldehyde dehydrogenases (ALDH) 1A1 and 3A1, but they are negative for fibroblast genes (like fibronectin, tenascin, and CD90/Thy1) and αSMA (for myofibroblasts) [6,13,14,15,16]. CSKs also express intracellular CHST6 (N-acetylglucosamine-6-O-sulfotransferase), an enzyme transferring sulfate residues to KS. Due to the presence of crystallins inside the cytoplasm of CSKs, the matching refractive index between cells and the surrounding stroma causes the invisible appearance of cells and minimal light scattering [17].

CSKs are also instrumental in stromal wound healing, which is a complex and highly orchestrated process that involves a series of cellular and molecular events aimed at restoring the structural integrity of the cornea following injury. However, this reparative response, while crucial for tissue recovery, can lead to the formation of scar tissue, potentially compromising corneal transparency and visual function [18]. CSKs retain the capacity to enter the cell cycle with a transition into a fibroblastic phenotype after stromal injury/trauma with the emergence of several cytokines, chemokines, growth factors, and conditions such as inflammatory response, fibrosis, and neovascularization (Figure 2). The recruitment of inflammatory cells to injured stroma is associated with the local synthesis and release of cytokines. This process is dependent on the severity and scale of injury. Mild corneal injury can be restored by the regeneration of CEpi, repair of basement membrane, and replenishment of keratocytes [19]. In case of severe injury, the release of profibrotic cytokines (such as transforming growth factor β, TGFβ, platelet-derived growth factor, PDGF) and chemokines (e.g., interleukin-1, IL-1, and tumor necrosis factor-α, TNFα) from the injured CEpi and ruptured basement membrane enter the stroma and activate the surviving CSKs to become repair-type stromal fibroblasts (SF). Stimulated by serum factors, chemokines, and cytokines (e.g., transforming growth factor-beta TGFβ, basic fibroblast growth factor bFGF, and platelet-derived growth factor PDGF), the cells mediate Smad- and non-Smad-dependent signaling pathways, such as phosphoinositide-3-kinase PI3K/Akt and JNK signaling. SFs express fibronectin receptors and produce and deposit repair-type ECM proteins (including fibronectin and SPARC) and collagenases to promote tissue remodeling. On the other hand, SFs lose keratocyte phenotypes and become migratory. They further produce cytokines, chemokines, metalloproteinases, and collagenases to attract infiltration of inflammatory cells (monocytes, macrophages, lymphocytes, and fibrocytes) to the site of injury. The presence of pro-inflammatory and chemotactic factors, such as interleukin-8, further stimulate SF generation and repopulation via the P-selectin pathway [20]. Some SFs transform into highly contractile myofibroblasts with a loss of proteoglycans and crystallins [6,21]. These cells express fibronectin receptors (α5β1 and αvβ3 integrins) that promote the assembly of fibronectin fibrils to conduct mechanical force during wound matrix contraction. Fibrosis involves SF and myofibroblasts that excessively produce abnormal ECM, which is deposited in a disorganized manner and with excessive tissue contraction, resulting in scar tissue formation. The scarring in the cornea changes light refraction and, in excessive volume, can impede the transmission of light rays. It is mostly irreversible due to the poor regeneration of CSKs, though a few reports have shown a reversal of fibrosis by cytokine action [22,23]. Currently, the most effective way to eliminate corneal scarring is corneal transplantation.

## 4. Etiology of Corneal Scarring Disorders

The multifaceted etiology of corneal stromal injury and wound healing encapsulates a broad spectrum of conditions, each contributing to the intricate mechanisms governing the reparative response of the cornea. Examining the distinct types of corneal disorders and their underlying causes is crucial for comprehending the complexities of the wound healing process and refining therapeutic strategies. This comprehensive understanding not only aids in mitigating the risk of undesirable outcomes but also provides a foundation for innovative interventions.

According to the World Health Organization (WHO), corneal blindness is responsible for ~5.1% of total blindness globally [2]. Holland et al. reported that about 10 million people suffer with bilateral corneal blindness worldwide [24]. Among them, about 2 million cases of blindness are caused by corneal ulceration and trauma [25]. In the United States, corneal disorders account for vision loss in nearly 4% of the population. Each year, approximately 2 million people suffer from corneal injury [26]. Different insults, including trauma, infection, chemical burns, ocular surgeries, and acquired and inherited ocular diseases, lead to corneal inflammation and fibrosis, eventually causing corneal blindness if the patients are not treated in a timely manner (Figure 3).

### 4.1. Ocular Trauma

Ocular trauma stands as a primary contributor to corneal stromal wounds, often arising from various sources such as sports-related incidents, workplace accidents, or domestic injuries. About 0.5 million people worldwide have blindness secondary to ocular trauma [27]. The impact can range from superficial corneal abrasions to more severe lacerations or perforations. The degree of trauma dictates the inflammatory response, with heightened cytokine release and immune cell infiltration initiating the reparative cascade. The subsequent fibrotic phase results in scar tissue formation, compromising the visual acuity. Wound site contamination after injury (e.g., agricultural accidents) gives an additional risk of corneal ulceration and vision loss [28]. For injury and scarring that involves the central cornea, the scar can worsen vision due to an altered curvature and irregular astigmatism. Rehabilitation requires contact lenses or excimer laser phototherapeutic keratectomy [29]. These can only be used for superficial scars, while deeper or full-thickness scars require some forms of corneal transplantation (keratoplasty).

### 4.2. Corneal Infection

Corneal infection represents another significant cause of ocular morbidity and blindness worldwide [30]. The infectious causes include bacterial, viral, fungal, and protozoa agents. Corneal ulcers resulting from infectious etiologies can lead to progressive stromal damage. Inflammatory responses triggered by pathogens can exacerbate tissue injury, adding layers of complexity to the wound healing process. Timely diagnosis and targeted antimicrobial interventions are crucial to curbing infection and minimizing the impact on corneal transparency. Depending on the severity, corneal scarring due to infection can persist even after the complete healing of the ulcer [31]. Corneal vascularization induced by infection can increase the risk of graft rejection post corneal transplantation [32]. On the other hand, peripheral ulcerative keratitis (PUK) is a form of noninfectious keratitis found in association with many systemic diseases [33]. After anterior uveitis, PUK is the second most common ocular complication of autoimmune disorders. Collagen vascular diseases account for 50% of all peripheral ulcerative keratitis cases with rheumatoid arthritis most commonly implicated [34]. Peripheral ulcerative keratitis also correlates with Wegener granulomatosis, relapsing polychondritis, polyarteritis nodosa, Churg–Strauss syndrome, and microscopic polyangiitis.

### 4.3. Chemical Injury

Chemical injury arising from the exposure to corrosive substances is a common cause of corneal haze and loss of corneal functions. This is due to a significant loss of CSKs and limbal stem cells [35]. The immune response from infiltrating inflammatory cells and neovascularization can be intensive [36]. The migration of conjunctival epithelium (conjunctivalization), due to a breakdown of the limbal barrier, and the formation of fibrovascular pannus pose unique challenges to visual prognosis and the response to therapy and rehabilitation. Meticulous management involves amniotic membrane grafting at the acute phase and limbal stem cell transplantation during the chronic phase with additional ocular surface reconstruction procedures [37,38]. The extremely dry ocular surface post chemical injury poses increased risk of corneal transplant rejection. Keratoprosthesis surgery may be required for visual rehabilitation [39].

### 4.4. Ocular Refractive Surgeries

Ocular refractive surgeries have become increasingly popular for treating refractive errors and astigmatism. Procedures such as Laser-Assisted In Situ Keratomileusis (LASIK), Photo-Refractive Keratoplasty (PRK), and Small Incision Lenticule Extraction (SMILE) remove or reshape corneal tissue to correct refractive errors [40]. While these surgeries are generally safe and effective, variations in individual healing responses can contribute to complications, including the formation of corneal haze and post-operative scarring, which cause suboptimal visual recovery. Undue activation of CSKs, fibroblast proliferation and migration, and excessive deposition of collagens and ECM remodeling have been identified to play significant roles in post-surgical scar formation [41,42].

### 4.5. Acquired and Inherited Corneal Disorders

Acquired and inherited corneal disorders represent additional contributors to corneal stromal abnormalities. Keratoconus (KC) is an ectatic corneal disorder characterized by progressive focal thinning, corneal steepening, and protrusion that leads to impaired vision [43,44]. The formation of an eccentric conical apex is the typical clinical presentation, whereas central scarring is seen in many advanced cases. Increasing evidence shows that KC has a multifactorial pathogenesis influenced by genetic, biomechanical, environmental, and behavioral factors [45]. Elevated inflammatory factors, e.g., TNFα, IL6, matrix metalloproteinases (MMPs), and reduced lysyl oxidase (LOX), have been reported by us and others in KC patients’ CEpi, Cst, and tears that affect the corneal stromal structure and organization [46,47,48].

Stevens–Johnson syndrome (SJS) is a type IV hypersensitivity immune-driven disorder with acute blistering in the skin and mucous membrane [49]. The ocular changes can range from minimal ocular surface involvement to severe scarring and blindness [50]. The management involves treatment of the associated dry eye and keratopathy, mucous membrane, or amniotic membrane grafting to target persistent epithelial defects and ocular surface inflammation. Owing to the poor surface stability, dry eyes, inflammation, and extreme xerosis, corneal transplantation can be suboptimal and graft rejection can occur.

Genetic corneal diseases affect corneal development, organization, and the cellular functions of one or more layers of corneas. Corneal dystrophies (CDs) predispose individuals to aberrant corneal changes with visual impairment due to dysfunctional corneal cells, lacrimation, and opacities [51]. Stromal homeostasis and health, function, and clarity are influenced by epithelial–stromal and stromal dystrophies, including lattice CD, granular CD types I and II, Reis–Buckler’s CD, Thiel–Behnke CD, macular CD, and Schnyder CD. Moreover, keratopathy is associated with different local or systemic disorders, like endocrine and inflammatory disorders. Neurotrophic keratopathy, stemming from the defective innervation to the cornea, is diagnosed with a loss of corneal sensation, impaired wound healing, persistent epithelial defects, and in severe cases corneal ulceration, melting, perforation, and scarring [52]. Diabetic keratopathy is prevalent among patients with systemic diabetic mellitus, featuring common clinical presentations, like CEpi erosion, superficial punctate keratopathy, reduced CEpi regeneration, and suppressed corneal sensitivity and visual acuity [53]. Various keratopathies exhibit abnormal arrangement of stromal ECM, leading to increased light scattering and opacities.

Overall, the etiology of corneal disorders encompasses a diverse array of factors, each with its unique challenges of correction and healing outcomes. From traumatic injuries and infections to ocular surface conditions and surgical interventions, understanding the specific context of the corneal insult is paramount for tailoring therapeutic strategies and optimizing visual outcomes. Ongoing research into the intricate mechanisms of corneal wound healing and scar management will continue to refine our understanding, providing a foundation for innovative approaches to effectively manage these diverse clinical scenarios.

## 5. Current Clinical Management of Corneal Scarring

The management of corneal scarring and stromal abnormalities requires a nuanced and multifaceted approach, incorporating a variety of pharmacological, biological, and surgical interventions. The choices depend on various factors, including the severity, depth of injury, and the underlying cause of disorder. Current treatment modalities aim to meticulously regulate the inflammatory response, promote appropriate wound healing, and curtail the development of scar tissue, yet they are not without challenges.

### 5.1. Topical Antibiotics 

Topical antibiotics remain a cornerstone in managing corneal stromal wounds, offering effective prophylaxis against microbial infections that can exacerbate abrasion damage. Agents such as fluoroquinolones are frequently employed, with their rapid onset of action and broad-spectrum coverage against bacteria [54,55]. The advantages of localized application and minimized systemic effects make antibiotics relatively accessible and simple to use. However, their overuse can lead to antibiotic resistance. Additionally, their penetration into the stromal layers may be limited, necessitating careful monitoring to prevent secondary infections and maintain the delicate balance of the ocular microbiome.

### 5.2. Topical Corticosteroids

Topical corticosteroids, such as prednisolone acetate, play a crucial role in suppressing inflammation and fibrosis by downregulating TGFβ-mediated events, mitigating immune responses, and alleviating discomfort [56]. The prevention of post-PRK haze and scarring was demonstrated by topical steroids and immunomodulatory agents like 0.03% tacrolimus and 0.05% cyclosporine [57]. However, their use necessitates careful consideration due to the delicate balance required between inflammation control and potential side effects, including delayed wound healing and increased risk of infection. Prolonged use of steroids can lead to adverse effects, including cataract formation and increased intraocular pressure. Ocular non-steroidal anti-inflammatory drugs (NSAIDs) are prescription medicine for prophylaxis and the treatment of ocular inflammation and/or pain associated with ocular conditions, usually in the post-operative setting. However, complications following topical NSAID use include delayed wound healing and an increased risk of corneal melting [58,59].

### 5.3. Mitomycin C (MMC) 

Mitomycin C (MMC) is a classical DNA-conjugating agent that forms covalent linkages within DNA strands, thereby inhibiting DNA replication and transcription and inducing irreversible senescence of cells [60]. This property has garnered attention for its potential in inhibiting fibroblast activity and preventing excessive scarring, particularly in cases of corneal haze following refractive surgeries [61,62]. This property is particularly advantageous in decelerating the development of stromal fibrosis, a common concern following refractive surgeries. The controlled application of MMC during surgery enables precise modulation of the wound healing process, demonstrating its potential as adjunctive therapy. However, it is important to note that the increased exposure to MMC may result in cytotoxicity, delayed wound healing, scleral calcification, ulceration, necrotizing scleritis, and damage to the corneal endothelium and ciliary body [63,64].

### 5.4. Amniotic Membrane (AM) Grafting 

Amniotic membrane (AM) grafting is a surgical procedure to suppress inflammation, promote corneal epithelialization, and reduce further complications, fostering an environment conducive to optimal wound healing [65]. In advanced corneal ulcer and large perforations, multi-layer AM grafting and tenon’s corneal patch grafts may be the choice of treatment [66]. The human AM is composed of a complex ECM containing collagen, laminin, fibronectin, and other matrix proteins. Our proteomic study showed that human AM possessed a suppressive activity on TGFβ/Smad signaling which governs the fibrosis gene expression [15]. AM also exhibits an anti-angiogenic property, limiting the formation of new blood vessels. This is particularly relevant in conditions where neovascularization compromises corneal clarity [67].

### 5.5. Collagen-Based Hydrogel

Collagen-based hydrogel emerges as a promising candidate for corneal wound healing, aiding in epithelial organization, preventing hypertrophic changes, supporting differentiation, and facilitating cell delivery and transplantation [68]. Its applications support wound healing and provide protection to other ocular tissues. The primary application in ophthalmology is collagen shields as post-operative bandages and as a delivery device [69]. BIO-Cor (from Bausch+Lomb Pharmaceuticals) slow-released hydrophilic drugs in the cornea and aqueous humor. Other drugs, such as prednisolone, cyclosporine, and ofloxacin, were successfully delivered to the anterior segment by this approach [70].

## 6. Emerging Therapeutic Strategies for Stromal Regeneration and Scar Inhibition

The main concern of patients with corneal scarring is the restoration of vision. Ongoing research efforts are focused on developing innovative methods to treat stromal defects and restore corneal clarity and functions. The field of regenerative therapy has emerged as a beacon of hope introducing novel approaches to address a spectrum of corneal opacities and scarring. This transformative branch of medicine seeks to harness the inherent reparative capabilities of cells to restore tissue function and integrity. In contrast to conventional treatments aimed at symptom management, regenerative therapies rejuvenate and restore the corneal cell populations and functions, offering the prospect of long-term tissue repair and functional restoration.

While stromal defects leading to opacities and scarring involve a loss of native stromal cells that function on stromal homeostasis, the restitution of healthy stromal cells depositing native stromal ECM proteins and collagens will improve the stromal matrix composition and its organization, recovering the biomechanics, refractivity, and clarity.

## 7. Cell-Based Approach for Corneal Wound Healing and Scar Management

### 7.1. Stromal Keratocytes as a Novel Therapeutic Tool for Scar Inhibition

Native CSKs are difficult to expand ex vivo, as they transit to fibroblasts when propagated with serum and growth factors in culture [18,71]. Our laboratory developed a robust method to expand *bona fide* CSKs (Figure 4A) [15], which specifically (1) produce and deposit stromal collagens, particularly type I collagen; (2) express stromal proteoglycans (such as keratocan, lumican, and decorin) and stromal crystallins (transketolase, aldehyde dehydrogenase 1A1 and 3A1); and (3) become quiescent in a serum-deprived condition and express CD34 and integrins on the cell surface for intercellular communication [9,15,72]. Using in vivo mouse and rat corneal stromal injury models (induced by excimer laser-mediated irregular photorefractive keratectomy and mechanical ablation by high-speed Algerbrush burring, respectively), CSKs were intrastromally injected to corneas with pre-existing early corneal haze and opacities. The treatment resulted in (1) a replenishment of native CSKs inside the injured stroma, (2) a reduction of corneal haze and improved corneal clarity, and (3) a recovery of native-like stromal ECM and collagen fibril organization [9,73]. In a manuscript under review, we further reported that the CSK-injected stroma had a recovery of stromal collagen fibrillar architecture (matrix ordering) revealed by small-angle X-ray scattering analysis, and fibrillar diameter and spacing under transmission electron microscopy. (4) This study further showed an improvement of visual functions in these CSK-treated rats. Using a survival test with the Morris water maze system, the rats travelled with a significantly shorter distance for survival and low escape latency, when compared to rats with non-treated injured corneas or treated with fibroblast injection. These results thus demonstrate that CSK treatment promotes stromal regeneration as it restores the stromal matrix composition and structural organization by depositing the right combination of ECM proteins. Inspired by these findings, this CSK injection strategy can be a suitable tool to correct or reverse stromal thinning and structural defects in corneal ectasia and keratoconus disorders.

Before administering CSKs to KC stroma, several unknowns must be clarified. Successful cell engraftment ensures sustained cell survival and functionality. (1) Can CSKs stably engraft in a KC stroma with degenerating the matrix structure and biomechanics (altered stiffness and viscoelasticity)? (2) Will the newly laid collagen fibrils align and configure similarly to that inside the native stroma? (3) Will the injected CSKs enhance the functions of the host stromal cells through cell–cell or paracrine activities to augment the stromal correction? This information will provide the insight to predict the therapeutic potential of CSKs in KC correction. It will also identify the mechano-transduction pathways of CSKs involved in sensing the 3D microenvironment and any potential regulators to improve cell engraftment in a biomechanically unfavorable environment.

The limited propagation of CSKs ex vivo potentially hinders the use of cells for clinical purposes. In our optimization study, primary CSK cultures were added with human amnion stromal extract to inhibit TGFβ-mediated profibrotic changes and cells expanded in a low level of serum condition were free of fibroblast marker expression [15]. This allows CSKs to propagate up to 20–30 doublings while maintaining the keratocyte phenotypes. Beyond this, the cells inevitably transit to express fibroblastic features. Hence, adult stem cells with reasonable proliferative potential and can differentiate into keratocytes would be appropriate for stromal tissue regeneration. In recent years, significant progress has been made in using stem cell treatments for corneal scarring and stromal defects.

### 7.2. Stem Cell Therapy for a Scarless Corneal Stromal Regeneration

Adult stem cells have limited proliferative potential and differentiation capacity to different cell lineages. In the last few years, human mesenchymal stem cells (MSC) from ocular and non-ocular sources have gained much interest in corneal stromal regeneration. Different studies, including reports from our group, have demonstrated that these human stem cells not only survived in a xenogenic condition of animal models, without inducing any immune and inflammatory responses and differentiated into mature keratocytes, but also exhibited keratocyte functions by depositing new stromal ECM components (collagens and proteoglycans) in the host stroma and remodeled the defective stroma to reduce scarring and improve corneal transparency [8,9,74,75,76,77,78,79,80,81].

Mesenchymal stem cells (MSCs) can be obtained from different human tissues, such as adipose, bone marrow, dental periodontal ligament and dental pulp, hair follicle, umbilical cord, and cornea (Figure 4C) [82]. Various MSC types have demonstrated induced differentiation capacity to adopt keratocyte features and expression of specific markers (including keratocan and lumican) [77,83,84,85] and were safe after administration into the host cornea due to their immunomodulatory properties [8,75,76,78,86]. The stromal remodeling effects of MSCs could be attributed to their secretion of paracrine factors, such as PEDF, HGF, and TGFβ3, which could improve the survival of local CSKs, inhibit cell apoptosis, and upregulate ECM protein expression, thus enhancing stromal repair and wound healing [87]. On the other hand, several MSC types are known to produce and secrete pro-angiogenic cytokines and growth factors, such as VEGF, HGF, FGF2, IL6, and IL-8, which come with the risk of neovascularization that can abrogate the corneal immune privilege and increase the rejective risk if a corneal transplant is performed [88]. Recently, clinical trial data were published for the treatment of KC patients’ corneas of advanced stages with autologous adipose-derived MSCs (ADSC) obtained by elective liposuction (Table 2) [89]. The cell suspension was administered to a mid-stroma femtosecond laser-assisted lamellar pocket of KC corneas. The treatment resulted in a higher stromal cell density, modulated scarring, and displayed neo-collagens, together with a moderate efficacy in terms of visual improvement (about two lines gain) [90]. The long-term observation up to 36 months did not detect changes of corneal keratometry readings and subjective refraction. However, the new ECM production did not quantitatively improve the thickness of the KC corneas, although the stromal cellularity was significantly increased within the anterior, mid, and posterior stroma [91].

Corneal stromal stem cells (CSSCs) isolated from the anterior limbal stroma or limbal-derived MSCs are recognized as the progenitors of CSKs at the central stroma (Figure 4B). Alongside the expression of typical MSC surface markers (CD73, 90, 105, 166, and STRO-1), these cells express other stem cell markers, including ABCG2, Pax6, Bmi-1, and Notch-1, and neural crest genes (Six2 and Six3) [92,93]. In vivo, human CSSCs show regenerative potential on the corneal stroma [94,95,96]. Using a Lumican knockout (Lum^−/−^) mouse model, Du et al. illustrated that an intrastromal injection of human CSSCs restored Lum expression and corrected the stromal matrix, yielding collagen fibrils with a uniform size and interfibrillar spacing [75]. This xenogenic transplantation was safe and without any risk of inflammation and rejection. The stromal regenerative and scar-inhibitory effects of CSSCs were demonstrated using murine models of anterior stromal scarring after mechanical debridement, alkali burns, or cold injury by liquid nitrogen [7,8,97]. The topical administration of human CSSCs in a fibrin gel vehicle successfully reduced stromal tissue inflammation via TSG-6 expression (less infiltration of CD11b^+^/Ly6G^+^ neutrophil and CD25^+^ macrophage) [97,98], downregulated fibrosis gene expression (including fibronectin, tenascin C, and αSMA), lowered opacity-related light scattering, and regenerated a native-like stromal matrix organization with uniform collagen fibril arrangement [7]. This rescue effect did not appear when the injured corneas were treated with vehicle only or with stromal fibroblasts. Our group recently reported that the stromal recovery by CSSCs was further improved by an additional CSK injection treatment, which strengthens the healing stroma with proper collagen types and stromal specific proteoglycans [9]. Building upon these numerous positive outcomes from pre-clinical studies, we established clinical-grade human CSSC manufacturing with GMP-compliant protocols [99]. The stem cell phenotypes and the characteristic capability to differentiate into CSKs were confirmed. In addition, we developed an important quality management system using in vitro quality control assays for primary CSSCs screening to identify their in vivo anti-scarring potency and effectiveness [99]. The calculation of the Scarring Index (SI) in correlation to the in vivo scar inhibitory outcome has demonstrated that CSSCs with SI < 10 had a predicted 50% scar reduction potency while cells with SI > 10 were ineffective to control scarring and should be excluded for patient use (International PCT/US23/27823).

## 8. Cell-Free Approach for Stromal Wound Healing and Regeneration

### 8.1. Extracellular Vesicles (EVs) 

Extracellular vesicles (EVs) are natural nano-sized (70–200 nm diameter) membrane-bound extracellular vesicles released by cells, and they carry genomic DNA fragments, different forms of RNAs and small RNAs, proteins, and lipids [100]. They are responsible for intercellular communications, hence mediating a trophic support to other cells and tissues. In the past decade, exosomes/EVs have emerged as a novel therapeutic tool for various systemic and non-systemic disorders. While tissue and cell-based therapies face different challenges, including immunogenicity, risk of rejection, cell stability, and homogeneity, exosomes demonstrate high consistency and stability, safety, ease of administration, and tunability of dosage and frequency according to the disease severity. Once exosomes are internalized by target cells via receptor-mediated endocytosis or membrane fusion, they release contents and regulate various signaling cascades. Direct head-to-head comparisons between EVs and their parental cells showed that exosomes are the mediator of MSC activity as both display comparable therapeutic activity [101,102,103]. MSC-derived exosomes (MSC-Exos) influence cell and tissue processes, such as differentiation, inflammation modulation, angiogenesis, and immunosuppression (Figure 5) [104,105]. In ocular applications, periocular or intravitreal injections of MSC-Exo/EVs reduced inflammation and improved visual function in animal models of uveitis, retinal injury, and diabetic retinopathy [106,107]. In the field of cornea, rabbit corneal stromal cells cultured with ADSC-derived exosomes exhibited robust proliferation, less apoptosis, and deposition of ECM collagen [108]. Administration of human umbilical cord MSC-derived exosomes carrying β-glucuronidase to transgenic mouse corneas with mucopolysaccharidosis resulted in a reduced accumulation of glycosaminoglycans and corneal haze development [109]. Such paracrine activity of MSCs also enhanced CSK survival by inhibiting apoptosis [87]. Additionally, the treatment of iPSC-MSC-derived exosomes to a rat corneal injury model promoted corneal epithelium and stromal reconstruction, leading to scar inhibition [105].

In a recent study by Ong et al., using EVs derived from a consistent source of ESC-derived MSCs for topical treatment to rat corneas with early scarring after irregular phototherapeutic keratectomy, the treated corneas showed significantly faster epithelial wound closure (*p* = 0.041), reduced haze levels (*p* = 0.002) and fibrosis (fibronectin and collagen 3A1 expression), and attenuated neovascularization (Figure 5) [110]. The EV-treated corneal tissues displayed a regenerative immune phenotype characterized by a higher infiltration of CD163^+^, CD206^+^ M2 macrophages over CD80^+^, CD86^+^ M1 macrophages (*p* = 0.023), reduced pro-inflammatory IL-1b, IL-8, and TNFα, and increased anti-inflammatory IL-10. Hence, MSC EVs alleviated corneal insult effects through anti-angiogenesis and immunomodulation towards a regenerative and anti-inflammatory phenotype. Shojaati et al. (2019) isolated EVs from human corneal stromal stem cells (CSSCs) and topically applied EVs in a fibrin gel to the corneal surface after stromal debridement injury [111]. After 2 weeks, the EV-treated corneas showed reduced opacities compared to untreated and vehicle-only injured controls. They showed that the EV treatment was effective in preventing neutrophil infiltration, reducing fibrosis marker expression, and recovery of the stromal organization. Our group recently characterized these CSSC-derived EVs using Nanostring microRNA profiler platform (Human v3 miRNA assay) and identified the presence of anti-fibrosis microRNAs hasa-miR-29a and miR-381-5p) in the Evs cargo [112]. In vitro, the expression of these two microRNAs reduced the lipopolysaccharide-stimulated M1 response of mouse macrophages and suppressed the TGFβ1-caused fibrotic reaction of human primary CSKs. Topical treatment of CSSC-EVs over-expressing both microRNAs inhibited corneal scar development in a mouse model of anterior stromal injury by Algerbrush debridement.

### 8.2. The Extracellular Matrix (ECM) 

The extracellular matrix (ECM) serves as a scaffolding for tissues and organs throughout the body, providing the structural and functional integrity [113]. It contains diverse components of matrix proteins, either structural (including collagens, elastin, fibronectin, laminins, tenascin) or non-structural (including integrins, growth factors, MMPs). It influences a wide range of cellular processes including adherence, migration, differentiation, signaling, and wound healing. As ECM remodeling is demonstrated to be involved in normal wound healing and scar development, modulating ECM-mediated biochemical and biomechanical pathways could be a novel approach to influence tissue scarring and be therapeutic for scar inhibition. In a study by Yin et al., the application of micro-sized particles processed from the ECM of lymph nodes to a rabbit lamellar keratectomy corneal injury model reduced corneal inflammation and fibrosis and promoted scar-reducing tissue repair [114]. This study demonstrated that an ECM microparticle treatment prevented rabbit keratocytes from transforming into myofibroblasts.

On the other hand, decellularized ECM guides wound healing progression by coordinating the cell phenotype and ECM protein production by modulating the M1 and M2 macrophage phenotypes, which release cytokines for cell homing and induce tissue remodeling. It also influences fibroblast and myofibroblast differentiation by modulating collagen production and induces angiogenesis by promoting endothelial cell migration [115]. Particularly, a decellularized ECM from the amniotic membrane exhibited increased wound healing efficiency in severe corneal injury, being characterized with a shorter healing time for CEpi and a faster recovery for stromal opacity and thickness, compared with the control eyes [116].

Table 1 gives an overview of cell-based and cell-free approaches showing corneal healing and scar inhibitory effects. The treatments with expanded CSKs, CSSCs, or MSCs from extraocular sources contribute to stromal regeneration and restore the corneal functions. Moreover, the application of EVs or exosomes from CSSCs and MSCs yielded similar scar-reducing wound healing outcomes.

Over the last decade, the advances of cell-based therapy have led to the initiation of clinical trials for corneal disorders involving opacities, e.g., dry eye disease, limbal stem cell deficiency, and corneal ectasia (keratoconus). Registered under the U.S. National Library of Medicine (ClinicalTrial.gov, assessed on 7 March 2024), more than 10 clinical trials are ongoing. At least five of them propose using MSC-based therapy, one using cultivated CSSCs, and one with MSC-derived exosomes to treat corneal pathologies (Table 2). From the available publications, the treatments were shown to be safe with no adverse effect and were effective in improving the corneal condition. These results warrant further large-scale multi-center clinical studies and should be conducted to confirm the treatment effectiveness.

## 9. Molecular Approach in Corneal Wound Healing

Matricellular proteins are soluble non-structural proteins inside the ECM and play roles in modulating cell and tissue functions by interacting with cell-surface receptors, proteases, hormones, and other ECM structural proteins, like collagens.

### 9.1. Hevin 

Hevin belongs to the secreted protein acidic and rich in cysteine (SPARC) family of matricellular proteins, and is known to regulate cell adhesion, proliferation, and migration [117]. In a study of corneal wound response after excimer laser-induced irrPTK, hevin knockout (hevin^−/−^) mice exhibited aberrant wound healing and had heightened light-scattering reflective particles in the corneas at 3–4 weeks post-injury [118]. Immunohistochemistry and Western blot analyses showcased an early surge of myofibroblasts and αSMA expression, indicating an accelerated inflammatory and fibrotic response compared to the wild-type corneas. Intriguingly, the administration of recombinant human hevin (rhHevin) mitigated these processes and reduced early corneal haze. These findings indicate the multifaceted properties of matricellular proteins in corneal biology and wound healing reactions, opening avenues for targeted interventions in corneal pathologies.

### 9.2. Krüppel-like Factor 4 (KLF4)

Krüppel-like factor 4 (KLF4), a zinc-finger transcription factor, regulates epithelial cell differentiation and homeostasis in diverse epithelial tissue [119]. In the mammalian corneas, KLF4 is abundantly expressed in the CEpi formation, playing an essential role in CEpi homeostasis through coordinating the apical and basal polarity of epithelial cells and suppressing epithelial–mesenchymal transition (EMT) [120]. Mouse corneas with conditional KLF4 knockdown showed reduced expression of CEpi markers (E-cadherin, cytokeratin 12, and claudin-3 and 4) whereas mesenchymal markers (vimentin, β-catenin) and EMT markers (Snail, Slug, Twsit-1 and 2) were upregulated [120]. Fujimoto et al. reported that human CEpi cells with KLF4 knockdown by siRNA approach had increased profibrotic gene expression [121]. In contrast, cells overexpressing KLF4 had epithelial gene expression, but not affecting the mesenchymal markers. TGFβ treatment on these KLF4-overexpressing CEpi cells had a reduced SMAD2/3 phosphorylation and nuclear translocation, compared to controls. This indicates that KLF4 can mitigate TGFβ-mediated corneal fibrosis via EMT suppression and blocking TGFβ/Smad signaling and nuclear SMAD localization.

### 9.3. Inhibitor of Differentiation 3 (Id3) 

Inhibitor of differentiation 3 (Id3) belongs to a family of regulatory dimeric bHLH transcription factors that bind to the E-box (CANNTG) sequences on the promoter region of target genes to regulate cellular growth, proliferation, and differentiation [122]. Its overexpression inhibited stromal fibroblast differentiation to myofibroblast under TGFβ induction in vitro. In a rabbit model of corneal scarring, localized Id3 overexpression by AAV5-mediated Id3 gene transduction inside the corneal stroma was demonstrated to be a safe practice and it significantly reduced pro-fibrotic marker expression (αSMA, fibronectin, collagen III) [123].

### 9.4. SMAD7 

SMAD7 is an inhibitory SMAD that negatively regulates the TGFβ/Smad pathway. It binds to the TGFβ receptor (TGFβR1) preventing SMAD2/3 phosphorylation and its interaction with SMAD2 and SMAD4, and this abrogates the nuclear SMAD localization and inhibits TGFβ-mediated fibrosis and EMT [124,125]. Downregulation of SMAD7 by siRNA targeting in human stromal fibroblasts induced αSMA positive myofibroblast generation and this effect was reversed by AAV5-SMAD7 transfection for protein overexpression [126]. In a rabbit corneal wound model, recombinant AAV5-SMAD7 gene therapy reduced corneal haze and profibrotic gene expression, and the corneas showed no signs of inflammation, redness, or ocular discharge, indicating the treatment safety. This pre-clinical result shows that SMAD7 protein therapy is safe and therapeutically efficient to inhibit corneal scarring.

### 9.5. Bone Morphogenic Protein 7 (BMP7)

Bone morphogenic protein 7 (BMP7) mediates SMAD-1/5/8 signaling, suppressing SMAD2 phosphorylation to counteract the pro-fibrotic effect of TGFβ/SMAD signaling. In a rabbit keratectomy model, the topical administration of recombinant BMP7 suppressed TGFβ-related corneal fibrosis, as observed by the reduced density of αSMA+ myofibroblasts [127]. Similar results of myofibroblast suppression and inhibition of fibrosis were observed for BMP7 overexpression via DNA-coated gold nanoparticles and AAV transfection, respectively [128,129].

### 9.6. Decorin 

Decorin is a small leucine-rich proteoglycan present inside the corneal stroma and binds to collagen fibrils regulating the fibrillar spacing to minimize light scattering and maintain corneal transparency [130]. Decorin binds TGFβ and sequesters it in the ECM, potentially inhibiting the pro-fibrotic TGFβ activity. In a mouse keratitis model, topical decorin downregulated αSMA and fibronectin expression, promoted wound healing, and reduced corneal opacities [131]. It also modulates the activity of cytokines and growth factors, such as VEGF and PDGF, in the process of corneal neovascularization and haze formation [132].

### 9.7. Regenerative Biomolecules and Immunomodulators to Route Scar-Forming Healing to Scar-Free Healing

The processes of wound response entails changes of the tissue microenvironment that involve a multitude of dynamic and interactive molecular and phenotypic events initiated after injury [133]. These pathways preferentially lead to scar tissue formation. Even though conventional treatments could alter these routes, the affected tissue still develops a similar final scarring phenotype. If “early intervention” is given soon after injury (e.g., to inhibit inflammation and fibrosis), it has the potential to “re-route” or “re-direct” the healing pathways towards a scar-reducing or scar-free phenotype. This strategy can be made possible by establishing high levels of anti-scarring or regenerative cytokines relative to the levels of the pro-scarring molecules. Specifically, during the initial inflammatory phase, the infiltrated neutrophils, monocytes, or activated macrophages secrete pro-fibrotic molecules, like PDGF, TGFβ1 and 2, IL-1, and TNFα, guiding the transition from the inflammatory phase to the fibrosis phase, and further manifesting with fibroblast and myofibroblast development to close the wound site by scarring. A timely resolution of the initial inflammatory phase could be desirable to untie its transition to the fibrotic phase. Pro-fibrotic TGFβ recruit histone deacetylase (HDAC) to remove the acetyl group from the lysine residues of histones H3 and H4 of the anti-inflammatory genes, enabling the transition to fibrosis.

#### 9.7.1. HDAC Inhibitor 

HDAC inhibitor (e.g., Trichostatin A) prevents histone H3 and H4 deacetylation, maintaining the anti-inflammatory gene activity. This concept has been verified by Trichostatin A treatment blocking the TGFβ-mediated transformation of stromal fibroblasts to myofibroblasts [134]. In vivo, topical Trichostatin A to rabbit corneas injured by excimer laser-mediated keratectomy reduced corneal haze formation. A similar HDAC inhibitor, suberoylanilide hydroxamic acid (SAHA, vorinostat), also attenuated the differentiation of equine fibroblasts to myofibroblasts and modulated MMP production in vitro [135].

#### 9.7.2. Members of TGFβ Family 

Members of TGFβ family can either activate or inhibit fibrosis, mechanistically acting through both canonical TGFβ/Smad and non-Smad pathways. TGFβ1 and β2 isoforms promote fibrosis whereas TGFβ3 inhibits fibrosis and drives scar-free healing effects [136]. Our research has demonstrated that human CSSCs (corneal stromal stem cells) produced TGFβ3 when the cells were applied to corneal wounds, hence reducing fibrosis gene expression and opacity formation [137]. CSSCs with TGFβ3 knockdown via the siRNA method lost this scar-reducing effect. Other studies reported similar findings that TGFβ3 stimulates non-fibrotic matrix production in corneas [138,139]. In mammals, fetal tissues possess higher levels of TGFβ3 relative to TGFβ1 isoform. The skin wound of embryos at early gestation stages heals without scarring and regenerates the native dermal matrix, while scar-forming healing happens in late gestation and afterward [140]. Similarly, postnatal oral mucosa expresses a high TGFβ3/β1 ratio, and it heals without scarring [141]. Therefore, methods to increase the TGFβ3/β1 ratio soon after injury could influence the healing process and direct to a scar-reducing or scarless pathway. Nonetheless, creating a high TGFβ3/β1 ratio in wound tissue is very unrealistic due to their short half-lives (about 50 min in culture condition) [142], and is expected to be even shorter in injured/inflamed environments with a low pH, possibly due to the damaged tissue and release of enzymes during phagocytosis. Hence, it is highly imperative to design a method that is capable of improving the drug bioavailability and maintaining pharmacological effects in a sustained manner inside the injured tissue. A recent report by Yang et al. developed a nanoformulation using ceria nanoparticles (NPs) of which the surface was integrated with poly(L-histidine) that can be responsive to the endogenous pH changes. Since the polypeptide forms positive charges in an acidic pH, this induces the solubility transition between naïve and injured microenvironments, and enables the NPs binding to the cellular membrane surface in a charge-independent manner promoting tissue permeability of the NPs [143]. Using a rat model of corneal alkali burn injury, the authors showed that these ceria NPs integrated with poly(L-histidine) delivered acetylcholine chloride and SB431542 (TGFβ receptor kinase inhibitor blocking TGFβ/Smad signaling) in a sustained-release manner, promoting wound repair and preventing scar formation.

#### 9.7.3. Losartan 

Losartan is known as an angiotensin-converting enzyme (ACE) II receptor antagonist (a hypertension drug) and an inhibitor of pro-fibrosis TGFβ signaling. By blocking the ACE receptor, losartan contributes to an anti-inflammatory milieu. This effect is particularly relevant in corneal stroma, where inflammation must be carefully regulated to facilitate optimal wound healing. It has shown promise in inhibiting myofibroblast generation in rabbit corneas after blast injury by irregular PTKs [144]. When losartan was used in conjunction with corticosteroids (prednisolone acetate), corneal opacity (area and intensity) was significantly reduced and repopulation of keratocytes was observed in the stromal wound area [145]. The therapeutic benefit of losartan was revealed in a clinical case of severe corneal haze after complicated LASIK [146]. After 4.5 months of topical losartan treatment, both uncorrected and corrected distance visual acuity were improved from 20/200 to 20/30 and from 20/30 to 20/25, respectively. Corneal haze was significantly reduced. Though losartan seems to be a promising drug to suppress fibrosis development, it should be explored further to confirm its efficacy in treating corneal scarring.

#### 9.7.4. Hepatocyte Growth Factor (HGF) 

Hepatocyte growth factor (HGF) reduces fibrosis in various organs [147,148]. Shukla et al. reported that HGF activated Smad7 (inhibitory Smad) to prevent Smad2 phosphorylation and nuclear translocation, hence inhibiting pro-fibrotic TGFβ signaling and reducing myofibroblast generation [149]. It also promotes apoptosis of myofibroblasts by inducing MMP to degrade fibrotic ECM that is the anchor of myofibroblasts [150]. In a mouse model of corneal injury, HGF treatment suppressed ocular inflammation and accelerated CEpi healing [151]. Another similar study showed MSC treatment restored the transparency of a wounded cornea via HGF production [152]. However, de Oliveira et al. reported no difference of fibrosis regulation when HGF was topically applied as eyedrops to a rabbit model of superficial corneal stromal injury by excimer laser-mediated PRK [153]. Recently, patients with corneal scarring are being recruited to a prospective phase I clinical trial using human recombinant dHGF (with 5 amino-acid deletion) (CSB-001) (ClinicalTrial.gov ID NCT06257355) (Table 2). All subjects are dosed with CSB-001 four times daily for 14 days. If subjects had scars resolved on day 7, the topical treatment was discontinued. Subject corneas were examined for safety and efficacy assessments (area, maximum depth, volume, density of scar, contrast sensitivity, and visual acuity).

#### 9.7.5. Lumikine 

Lumikine on TGFβR signaling in treating corneal scarring. Lumican, a small leucine-rich proteoglycan, is a component of ECM and functions as a matrikine regulating various stromal cell activities (e.g., growth, migration, and gene expression of CSKs) and collagen fibrillogenesis and organization [154]. It also modulates the corneal inflammatory response via Fas–Fas ligand signaling [155]. Lumican binds to the Alk5 domain of TGFβ-activated tetrameric TGFβ receptors (TGFβR1). Kao’s group has discovered that a short peptide of thirteen C-terminal amino acids of lumican (LumC13) is essential for promoting corneal wound healing and CEpi cell growth and migration [156]. Lumikine, a stable derivative of LumC13 with a single amino acid substitution, was effective to suppress stromal scar tissue formation in mouse corneas after mechanical injury [157]. This finding shows that Lumikine can be a potential drug for corneal wound and scarring management. Despite this potential, the long-term safety and specific mechanisms of action are still under investigation. Precision in dosing and understanding individual variations in response are essential for their effective and safe utilization.

An overview of expressing target genes/proteins with the healing effect on corneal fibrosis and scarring is summarized in Table 3. Overexpressing certain genes could have potential risks and side-effects, especially overloading the cellular machinery of protein biosynthesis and quality control, causing misfolding, mis-trafficking, and post-translational problems [158]. These side effects could result in abnormal complex formation and cellular toxicity. Hence, further studies are warranted to investigate the effectiveness and safety in pre-clinical and clinical conditions.

## 10. Targeted Gene Silencing to Prevent Corneal Scarring

In corneal wound responses, a number of genes are upregulated in association to a scarring signaling cascade, including semaphorin 3A (SEMA3A), ubiquitin-specific protease-10 (USP-10), and calmodulin/Ca^++^-activated K^+^ channel 3.1 (Kca3.1). Their induced expression contributes to corneal fibrosis and scarring. Hence, the strategy of targeting these genes using siRNA can downregulate the fibrosis development and may be effective to prevent corneal scarring.

After corneal epithelial–stromal injury, the EGF released from the healing CEpi cells enters the stromal region and induces SEMA3A expression in stromal fibroblasts. SEMA3A, in combination with TGFβ, promotes fibrotic gene expression [159,160]. Hence, SEMA3A downregulation in stromal fibroblasts by siRNA targeting could represent a novel anti-fibrosis strategy.

USP10 is upregulated due to the cellular stress associated with wound healing. It binds to the nuclear p53 to promote apoptosis of keratocytes and epithelial cells, allowing neutrophils and macrophage infiltration [161]. Targeted silencing of USP10 by siRNA approach in wounded porcine corneas improved the stromal ECM arrangement, suppressed fibrosis gene expression (αSMA and fibronectin), and reduced immune cell infiltration [162]. These findings support the molecular targeting of USP10 for a scarless corneal wound healing.

Fibrosis is commonly associated with Ca^++^ signaling. Stromal keratocyte transition to fibroblasts and myofibroblasts is activated by cell polarization, which is associated with Ca^++^ influx and K^+^ efflux via the Ca^++^-activated K^+^ ion channels [163,164]. Accumulated KCa3.1 was observed on the surface of activated keratocytes, contributing to fibrosis [165]. KCa3.1 knockout mice showed reduced corneal haze formation and lower levels of fibrosis gene expression (αSMA).

In an ex vivo model of excimer-ablated rabbit corneas, a triple combination of siRNAs targeting scarring genes TGFΒ1, TGFΒR2, and CTGF significantly reduced haze levels by 55% and 68%, respectively, along with decreasing αSMA mRNA and protein levels. In contrast, haze-like scarring was observed in placebo-treated corneas, with elevated pro-fibrotic gene expression [166].

An overview of target gene silencing with the potential to treat corneal fibrosis and scarring is summarized in Table 4. Potential risks and side effects of knocking down specific genes are illustrated. Setting up this approach in corneal cells and tissue in vivo could be limited by the short-term effect of siRNA-mediated gene silencing, siRNA stability, and low efficiency, particularly in primary cells. In addition, gene alteration could lead to cell phenotypic variations and altered cellular signaling. Hence, further validation of their effectiveness and safety in pre-clinical and clinical studies is required.

## 11. Tissue Engineering Approach for Corneal Regeneration

Corneal tissue transplantation remains the most effective therapeutic option for replacing scarred tissues of the patient’s cornea and restoring the eyesight. However, this approach faces different disadvantages, including the risk of allograft rejection and the limited worldwide availability of transplantable donor materials. On average, 1 out of 70 patients can have access to the corneal transplant and this situation is even worse in developing and underdeveloped countries [167]. Hence, there is an overwhelming need for a transplantable device as an alternative option to rescue corneal blindness. The development of stromal tissue analogs with strong biomechanics and biocompatibility, safety, and clarity will provide a promising and cutting-edge strategy in the realm of corneal replacement.

### 11.1. Stromal Lenticule Engineering

This method involves the utilization of biological stromal/refractive lenticules, typically sourced from donor corneas after SMall Incision Lenticule Extraction (SMILE) procedures. The precisely cut discs of native stromal tissue are ultrathin, transparent, avascular, and mechanically robust with a well-organized collagen-rich ECM composition, and they are usually obtained from young and healthy corneas for the purpose of refractive corrections [168]. This native bioscaffold material has great potential for tissue repair and wound healing, and the tissue addition process to improve tissue strength and integrity. Lenticule implantation has been proven to be an effective treatment for hyperopia and corneal ectasia [169,170,171]. The lenticules can serve as carriers for cell delivery and the versatility of this approach allows for customization based on the specific needs of patient and the nature of corneal disorders. Lenticules can be customized for the purpose of reimplantation. This ultrathin tissue (usually 30–140 μm thick, depending on the diopter correction) were successfully thinned and reshaped using excimer laser ablation under controlled dehydration procedures [172]. Decellularization using different approaches with detergents and nucleases was shown to completely remove the cellular materials and antigenic molecules while retaining the structural and functional features of the ECM components, hence reducing the risk of immunogenicity and host rejection after reimplantation [173,174]. These preparations give additional advantages of using lenticules for tailored therapeutic interventions and enhance the likelihood of successful integration and functional restoration. Our recent study further reported the recellularization and reinnervation in these lenticule structures [175]. This is extremely beneficial for the implanted lenticules with the host tissue. A regenerative effect could be achieved by long-term quiescent CSK infiltration and repopulation, which can promote stromal collagen turnover and tissue remodeling. Overall, the lenticule engineering approach holds promise as a new avenue of corneal regenerative medicine. Its potential applications could extend beyond corneas, encompassing other tissue organs such as the skin and tendons.

### 11.2. Synthetic Non-Collagen-Based Scaffolds

This synthetic approach using non-collagen materials can provide a wide range of tunable mechanical strengths that can serve as good substrates for easy handling and graft delivery. However, the biocompatibility and long-term graft survival in vivo will not be comparable with collagen-based scaffolds.

#### 11.2.1. Gelatin-Based Hydrogels 

Gelatin-based hydrogels have emerged as a powerful tool in fostering controlled and efficient tissue regeneration. Gelatin, which is derived through the hydrolysis of collagens (denatured form of collagen), is more prone to biodegradation and absorption than collagen, and this is advantageous for avoiding long-term biological reactions that may induce opacification in certain tissue systems, such as corneas. Using a 3D culture system, Mimura et al. showed stromal fibroblast culture on gelatin hydrogen with the production of new ECM proteins [176]. Modifications by cross-linking collagen molecules to the gelatin hydrogel improved mechanical strength and Young’s moduli, with higher hydrophilicity for cell adherence, and better optical properties [177]. Other tunable physical and biochemical properties of hydrogels, such as stiffness and viscoelasticity, allow for the creation of an environment conducive to cell proliferation, thereby promoting tissue regeneration [178]. To minimize toxicity, like from crosslinking agents, various strategies have been developed for functional hydrogel preparation. These include photopolymerization, enzyme-enabled crosslinking, click chemistry using thiolene radical reaction, Diels–Alder reaction or azide–alkyne cycloaddition, and Schiff-base reaction.

Methacrylation of gelatin (GelMA), followed by photo-crosslinking with various wavelengths of light and photo-initiators, provides a greater control over crosslinking density and hydrogel porosity. The elastic modulus can be varied by changing the polymer concentration and light exposure time or employing a combination of crosslinking strategies. Incorporating both physical and UV crosslinking increases the mechanical strength, showcasing the potential of hybrid crosslinked GelMA hydrogels [179]. The rate of biodegradation for these hybrid crosslinked GelMA hydrogels was slower than that of only UV crosslinked GelMA. This tunability in degradation rate aligns with the desirable attribute of matching the healing cascade of corneal injuries or diseases. Photo-crosslinked GelMA can be effectively 3D-printed to mimic corneal stroma tissues, enhancing structure and scalability [180]. Figure 6 illustrates the bioengineering of a biomimetic corneal stroma with an incorporation of native stromal cells (keratocytes) in a GelMA which can be transplanted to the corneal defect and photocured in situ. Using this approach, a recent study by Huang et al. showed that the biomimetic corneal stroma restored the corneal structure and remodeled the stromal environment by proteoglycan secretion to promote transparency and inhibition of the inflammatory response to reduce stromal fibrosis and scar formation [181]. Composite hydrogels, such as hyaluronic acid-modified GelMA, also allows for the 3D printing of corneal scaffolds. In vivo study showed that this 3D-printed scaffold provided cues guiding stromal cells toward the directional and spatial organization and facilitated the ECM remodeling.

#### 11.2.2. Silk Fibroin

Silk fibroin, a natural biopolymer extracted from *Bombyx mori* silk cocoons, has emerged as a promising material for corneal scaffold fabrication and tissue engineering. Comprised of heavy and light chains, silk fibroin proteins can be genetically engineered and synthesized in the laboratory, allowing for versatility of the material design. These proteins can also be combined with other peptide sequences, further enhancing their adaptability. Regenerated silk fibroin can undergo modifications to optimize its functionality, including alterations to carboxylic and amide groups along the protein backbone. With the properties of high transparency, easy to model, controllable degradation, non-immunogenic, and with optimal mechanical resistance, this biopolymer has been reported to have a wide application in corneal tissue engineering [182,183]. Incorporation of corneal stromal stem cells (CSSCs) in multi-lamellar silk film architecture produced a 3D functional corneal stromal equivalent, which successfully reconstructed the corneal stroma in a rabbit model [184]. However, challenges arise in the application of unmodified regenerated silk fibroin for corneal tissue engineering due to the slow formation of nanocrystalline domains, impacting transparency and mechanical properties. Ongoing research endeavors to address this limitation by preventing the formation of nanocrystalline domains or disrupting the phase separation during β-sheet formation within the materials. A promising direction involves the development of modified regenerated silk fibroin–hyaluronic acid and composite hydrogels, showcasing the potential for applications as vitreous humor substitutes [185]. These advancements underscore the ongoing efforts to enhance the efficacy and applicability of silk fibroin-based materials in corneal tissue engineering and beyond.

#### 11.2.3. Chitosan 

Chitosan is a natural polymer derived from the deacetylation of chitin (a linear polymer of N-acetylglucosamine) [186]. It is the primary component of an exoskeleton of crustacean sources. The interest in chitosan and chitin relies on the myriad biological and technological properties—mucoadhesive, anti-inflammatory, antioxidant, anti-microbial, anti-fungal, anti-hyperglycemic, anti-tumoral, and wound healing. Together with its biocompatibility and degradability, chitosan can be suitable for corneal regeneration. It was reported to be used as a cell carrier for ocular surface and corneal endothelium [187]. When incorporated with other molecules, like collagen or silk fibroin, the product can serve as a stromal equivalent with increased biomechanical strength and for cell adherence and growth [188,189].

## 12. Summary and Future Perspectives

After injury or in various diseases, corneal stromal pathologies encompass inflammation, fibrosis, opacity formation, neovascularization, and stromal degeneration (such as keratoconus). These conditions often result in scarring and corneal deformation, leading to increased light scattering and blockage of light passage and astigmatism, which are the causes of corneal blindness worldwide. Current pharmacological treatments (including antibiotics, lubricants, steroids, mitomycin C) are able to treat mild opacities and delay their progression. However, the standard of care for moderate to advanced corneal blindness and established scarring relies on the surgical transplantation using donor cornea tissues (penetrating and lamellar keratoplasties) to replace the scarred tissues and restore corneal functions and vision. Nevertheless, the widespread use of surgery is limited by various factors, including the limited global supply of transplantable donor materials. Hence, more research directed towards an understanding of mechanisms and responses after injury or in the corneal tissues with specific gene defects and pathways promoting stromal repair/regeneration, scar reduction, and restoring corneal transparency is highly demanded. If the density of opacities can be reduced, other non-surgical therapies, such as the use of contact lenses, can correct corneal astigmatism and provide visual rehabilitation, thus lessening the demand for donor grafts.

In the last two decades, the development of autologous limbal stem cell transplantation has been a great success in restoring CEpi from persistent epithelial defects due to limbal failure and immunological problems. It can be anticipated that, in the near future, clinical cell therapy can be extended to corneal stroma (using keratocytes, stromal stem cells or MSCs) and corneal endothelium (using mature CEndo cells or progenitors). To achieve this, reliable and cost-efficient good manufacturing protocol (GMP)-compliant procedures, release quality control, and delivery techniques have to be established in cooperation with the regulatory authorities. Further work on treatment efficiency, stability, therapeutic outputs as well as ethical issues need to be clarified to facilitate the application of cell-based treatment in humans.

In the cell-free approach, the targeted modulation of wound response pathways or stages in the healing cascade represents a contemporary method for treating corneal scarring and opacities. The strategy of exosome or extracellular vesicle application holds potential for transferring therapeutic molecules (immunomodulators, anti-inflammatory/fibrosis cytokines, microRNAs, and mRNAs) to the target cell (surviving CSKs). This activity can prevent their transition to repair-type fibroblasts and scar-forming myofibroblasts by either preventing neutrophil infiltration, reduced apoptosis, or blocking the TGFβ pro-fibrotic signaling cascade. However, there are significant considerations with this strategy. The main concerns are the need for the up-scale harvest and isolation of exosomes, and the proper delivery of exosomes to the right type of target cells inside the injured tissues, as the activated macrophages engulf most of the exosomes administered. A system of slow and sustained release of exosomes, such as the encapsulation in hydrogel or incorporation with nanoparticles, should be explored. Additionally, enriching therapeutic materials within the cargo content of exosomes requires further consideration to enhance their efficacy in treating corneal scarring and opacities.

Another approach to addressing fibrosis involves redirecting the pathways from pro-fibrotic healing and scarring to a more sophisticated mechanism with reduced scarring or scar-free healing. This method can be achieved by regenerative molecules and immunomodulators. The high TGFβ3/β1 ratio observed in early embryonic and in mucosal tissues facilitates healing without scarring. Clinical treatment with Losartan blocks myofibroblast generation, supporting its therapeutic benefit as a topical drug for corneal injury. However, it requires frequent applications (six times a day for one to several months depending on the injury scale), which can be inconvenient for patients and can lead to poor compliance. Therefore, a convenient and effective drug delivery method should be determined.

Researchers have also delved into the emerging field of scaffold-based engineering to develop novel strategies for corneal wound healing, cell and drug delivery, and stromal regeneration. With the increasing popularity of SMILE procedures for refractive corrections, the extracted stromal lenticules with their native collagen-rich composition, strong mechanical strength, and transparency present an opportunity for treatment instead of being disposed as medical waste. The success in reshaping and decellularizing lenticules has paved the way to get high-quality transplantation-worthy stromal ECM scaffolds suitable for therapeutic use and regenerative medicine.

Translating these discoveries for clinical use could potentially address the current challenges of a global shortage of donor materials. However, as new data emerge, there is a need for a deeper understanding about corneal wound response and tissue healing, modulation of wound specific cellular and ECM changes, and control of stromal fibrosis processes. This will contribute to the development of novel therapeutics or repurposing drugs that leads to effective methods to alleviate stromal scarring and restore corneal functions and transparency. Ultimately this will aid in the clinical management and treatment of corneal blindness.

**Table 1 biomedicines-12-00649-t001:** Cell-based and cell-free approaches to modulate corneal fibrosis and scarring.

Approach	Types	Mechanisms of Action	Risks/Potential Side Effects	Limitations
Cell-based	Corneal stromal keratocytes	Produce and deposit native stromal collagen and proteoglycans to restore ECM composition	Transit to fibroblasts and MyoF under wound conditions, need to apply after pro-inflammatory and fibrotic cytokines are suppressed [9]	Low cell yield due to slow ex vivo expansion [15]
	Corneal stromal stem cells	Anti-inflammatory with TSG-6 expression; anti-fibrosis with TGFβ3 expression; differentiation to keratocytes [98,137]	Cell fate and phenotypic variation in response to pH changes and inflammatory response in corneal wound	Donor to donor variation in cell characteristics and functions [99]
	Mesenchymal stem cells from adipose, bone marrow	Anti-inflammatory; immuno-modulatory; keratocyte differentiation [76,78,89]	Uncertainty in ECM production specific to corneal stroma;risk of angiogenesis [88]	Donor to donor variation in cell features
Cell-free	Extracellular vesicles from CSSCs, MSCs	Anti-fibrosis microRNAs (miR19a, 29a, 381) to prevent M1 macrophage activation, suppress JNK fibrotic and TGFβ pathways [110,111,112,190]	Easy application with minimal immunogenic effects. However, uncharacterized EV content results in unwanted effects.	Large-scale cell culture to prepare EVs; clearance or binding of EVs to ECM restricts cellular uptake [191]
	Extracellular matrix	ECM microparticles reduced inflammatory and fibrotic gene expression; prevented MyoF generation [114]	Wide range of applications in different physical forms—sheets, suspension; easy to modify and functionalize	Material heterogeneity; need to develop isolation methods with high yield and purity [192]

**Table 2 biomedicines-12-00649-t002:** Clinical trials with corneal scarring treated by cell-based and cell-free approaches.

ClinicalTrial.gov ID Year Initiated	Title	Target Disease, Treatment	Sponsor	Status	Publications
NCT01562002 2012	Safety Study of Stem Cell Transplant to Treat Limbus Insufficiency Syndrome Phase I/II, double masked	LSCD	Institute of Applied Ophthalmobio-logy, Spain	Completed recruitment; N = 17 patients; No adverse effects. Improved CEpi healing	[193]
		Allogenic bone marrow MSCs Stem cells with amniotic membrane transplant			
NCT02291770 2015	Mesenchymal stromal cells treatment attenuates dry eye in patients with chronic graft-versus-host disease	GVHD-DED	Guangdong Provincial People’s Hospital, China	No adverse effects. In total, 12 out of 22 patients had improved dry eye score, ocular surface disease index scores, and Schirmer test results	[194,195]
		Allogenic bone marrow MSCs Intravenous injection			
	Phase III, multi-center, randomized, Open-label				
NCT02592330 2015	Limbal Stem Cell Deficiency (LSCD) Treatment With Cultivated Stem Cell (CALEC) Graft	LSCD	Massachusetts Eye and Ear Infirmary, USA	Completed recruitment	
		Cultivated autologous limbal epithelial cell graft			
	Phase I/II, open-label	CALEC Transplant			
NCT03687632 2018	ST266 Eye Drops for the Treatment of Persistent Corneal Epithelial Defects Phase II, multi-center, open-label	PED	Noveome Biotherapeutics	No adverse effects. A total of 10 out of 12 eyes had reduced PED area	[196]
		Multi-cytokine biologic solution from Amnion-derived Multipotent Progenitor culture Eye drops			
NCT03878628 2019	Treatment With Allogeneic Adipose-derived MSC in Patients With Aqueous Deficient Dry Eye Disease (MESADDE)	DED Kerato-Conjunctivitis Sicca Aqueous Tear Deficiency	Rigshospitalet, Denmark	No adverse effects. Decreased mean OSDI score, tear osmolarity; increased TBUT, Schirmer’s I test	[197]
		Allogeneic adipose-derived MSC			
	Early Phase I, open-label	Transconjunctival injection			
NCT04213248 2019	Effect of UMSCs Derived Exosomes on Dry Eye in Patients With cGVHD Phase I/II, open-label	Dry Eye	Zhongshan Ophthalmic Center, Sun Yat-sen University, China	Recruiting No adverse effects; reduced fluorescein scores, longer tear-film breakup time; increased tear secretion; and lower OSDI scores	[198]
		Umbilical MSC-derived exosomes Eye drops			
NCT04932629 2021	To Evaluate the Clinical Safety and Efficacy of Limbal Stem Cell for Treatment of Superficial Corneal Pathologies	Corneal scar and opacities	L.V. Prasad Eye Institute, India		
		Ex vivo cultivated allogeneic limbal stromal stem cells			
	Early phase I open-label	Topical with fibrin glue			
NCT05279157 2022	Autologous Adipose-Derived Adult Stem Cell Implantation for Corneal Diseases (ADASCs-CT-CD) Phase II	Corneal dystrophy, keratoconus	Vissum, Instituto Oftalmológico de Alicante, Spain	Completed No adverse effects; improved stromal cell density, modulated scarring, visual improvement (~2 lines gain)	[81,90,199]
		Autologous adipose MSCs Corneal implantation			
NCT06257355 2024	Study to Evaluate the Safety and Efficacy of CSB-001 Ophthalmic Solution 0.1% in Subjects With Corneal Scars	Corneal scar	Claris Biotherapeutics, Inc.	Recruiting	
		Human recombinant dHGF (hepatocyte growth factor) Eye drops			
	Phase I Open-label				

Note: LSCD—limbal stem cell deficiency; GVHD-DED—graft-versus-host disease–dry eye disorders; MSC—mesenchymal stem cells; TBUT—tear breakup time; OSDI—ocular surface disease index; PED—persistent epithelial defect.

**Table 3 biomedicines-12-00649-t003:** Target gene overexpression to modulate corneal fibrosis and scarring and their potential adverse effects and limitations.

Genes	Mechanisms of Action	Risks/Potential Side Effects	Limitations of Approach
Hevin	Suppressed early fibrosis; reduced myoF [118,200]	Not studied	Overexpression or misexpression of genes can induce phenotypic variations and extra stress of cells. Overloading of translational and protein biosynthesis machinery leading to folding, localization, degradation, and post-translational problems. Abnormal complex formation; cellular toxicity [201,202]
KLF4	Suppressed EMT and fibroblast activation; reduced SMAD2/3 phosphorylation	Negatively regulates cellular anti-viral immune response; complex effects on tumor inhibition; promotes pre-cancerous lesions [203,204]	
Id3	Suppressed MyoF generation	Positively suppressed TGFβ-induced IOP elevation; relates to oncogenesis but with exceptions [205,206]	
SMAD7	Reduced SMAD2/3 phosphorylation and inhibited EMT; restrained MyoF generation	Targets TGFβ receptor for proteasomal degradation; activates EGFR-signaling in carcinogenesis [207,208]	
BMP7	Suppressed pro-fibrotic TGF-β/SMAD signaling and pro-inflammatory cytokine production	Risk of cancer metastasis [209,210]	
Decorin	Sequestered TGFβ from receptor binding and suppressed fibrosis [130,211]	Altered proteoglycan content may modulate growth factor activity [130,211]	
HDAC inhibitor	Inhibited histone H3 and H4 deacetylation to modulate cell growth and differentiation, suppressing fibroblast and MyoF generation.	Multiple HDACs induce opposite effects on a single event, indicating the pan-inhibitory action of HDAC inhibitor could result in unwanted effects [212]	
Losartan	Blocked TGFβ signaling to suppress MyoF generation and fibrosis	A well-tolerated medication with few side effects [213]	
HGF	Activated Smad7 to inhibit TGFβ/Smad pro-fibrotic signaling and reduced myofibroblast generation; anti-inflammatory	Pro-angiogenic activity could lead to neovascularization; HGF/c-Met signaling to trigger tumorigenesis [214]	

**Table 4 biomedicines-12-00649-t004:** Target gene silencing or downregulation to modulate corneal fibrosis and scarring, and their potential adverse effects and limitations.

Genes	Mechanism of Action	Risks/Potential Side Effects	Limitations of Approach
SEMA3A	siRNA-mediated downregulation of fibroblast/TGFβ-fibrotic pathways	Neuron polarization defects; corneal sensory alterations; risk of VEGF-mediated corneal neovascularization [215,216]	Variable knockdown efficiency by siRNAs and instability and degradation of siRNAs inside target cells. Lack of reliable delivery methods—transfection approach is poor for primary cells and electroporation induces cell death. Altered target gene expression induces phenotypic variations and altered cellular signaling [217].
USP-10	siRNA-mediated downregulation of immune cell infiltration and fibrosis gene expression	Altered de-ubiquitination modulates multiple cellular issues, e.g., protein stability [202]	
KCa3.1	Using TRAM 34, an ion channel block to modulate Ca^++^-activated K^+^ signaling in fibroblast and MyoF activation; suppressed macrophages polarization towards M1 phenotype [165,218,219]	Affects cell growth and survival; triggers cell death [165,218,219]	

## Figures and Tables

**Figure 1 biomedicines-12-00649-f001:**
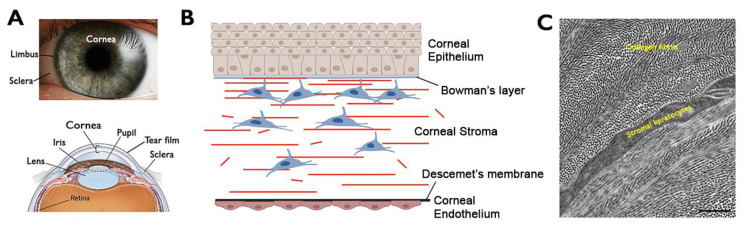
Human cornea structure and composition. (**A**) Human cornea anatomy. (**B**) The human cornea consists of five known layers—three cellular (epithelium, stroma, and endothelium) and two interface non-cellular layers (Bowman’s layer and Descemet’s membrane). (**C**) The corneal stroma is composed of the extracellular matrix with collagen fibrils organized as stromal lamellae which run orthogonally to each other. Stromal keratocytes are located between stromal lamellae. Transmission electron micrograph with scale bar 2 μm. Created with BioRender.com under license WD26KLB6G7, assessed on 11 March 2024.

**Figure 2 biomedicines-12-00649-f002:**
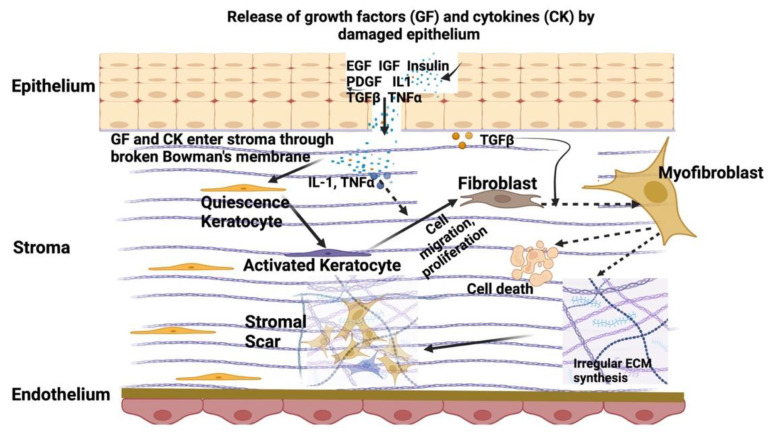
A schematic diagram elucidating corneal epithelial–stromal injury and stromal scar formation. In a corneal wound, the damaged epithelium triggers epithelial healing (cell migration and differentiation). The disrupted Bowman’s layer and the infiltration of neutrophils and macrophages allow the invasion of pro-fibrotic growth factors (GFs) and cytokines (CKs) into the stroma. This causes the surviving keratocytes to activate and transit into repair type stromal fibroblasts and contractile myofibroblasts, overproducing ECM proteins with disorganized arrangement and resulting in scar formation. Created with BioRender.com under license JX26IWLDIY, assessed on 1 March 2024.

**Figure 3 biomedicines-12-00649-f003:**
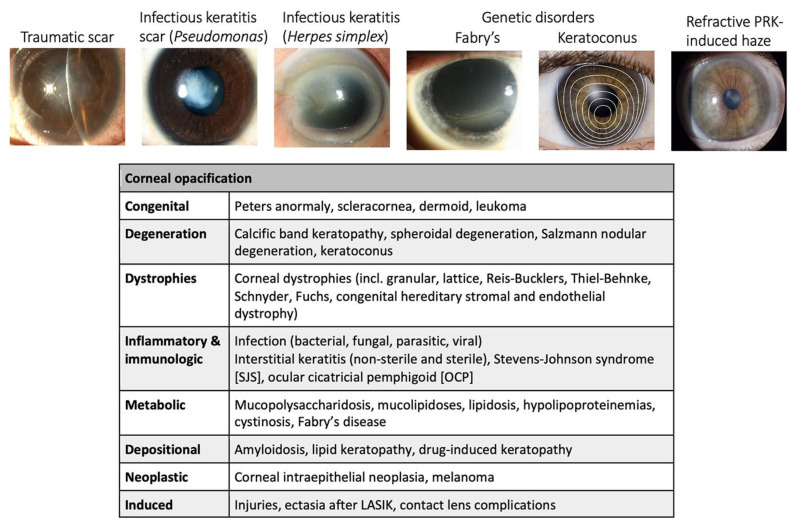
Causes of corneal haze and opacities.

**Figure 4 biomedicines-12-00649-f004:**
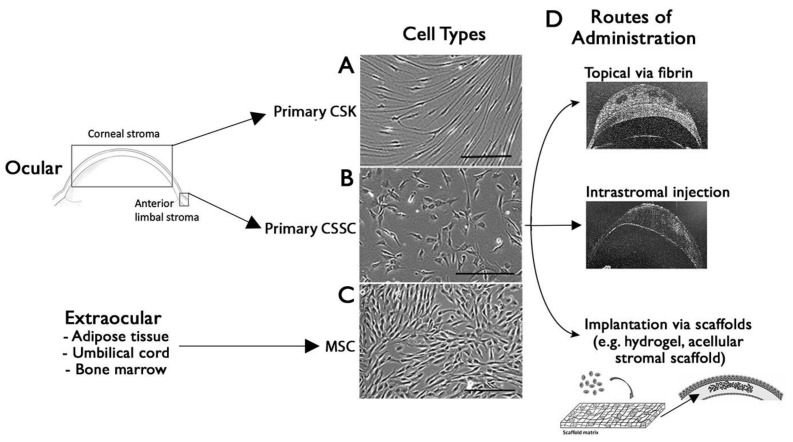
A schematic diagram of tissue sources of different cell types for corneal stromal regeneration and cell treatment modalities. (**A**) From ocular corneal tissue, the central stroma derives primary corneal stromal keratocytes (CSKs) for ex vivo culture (Yam 2018 Cell Transplantation). Propagated cells are induced to generate growth-arrested keratocytes expressing different keratocyte-specific gene markers and phenotypes. (**B**) The anterior limbal stroma derives primary corneal stromal stem cells (CSSCs). The ex vivo expanded cells express various stem cell and MSC gene markers, anti-inflammatory genes, and are capable of generating keratocytes. (**C**) Extraocular tissues, like adipose, umbilical cord, and bone marrow, are the sources of mesenchymal stem cells (MSCs) with multipotent differentiation potential. (**D**) Cells are administered to the cornea via topical application in a fibrin gel, intrastromal injection of cell suspension, and intrastromal implantation of cell-ladened tissue or hydrogel scaffolds. Scale bars: 100 μm.

**Figure 5 biomedicines-12-00649-f005:**
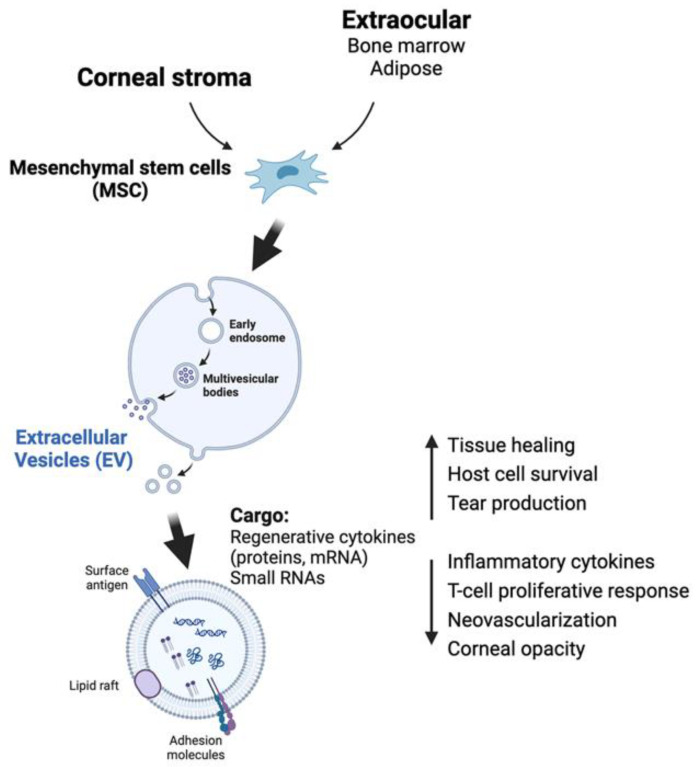
A schematic diagram depicting extracellular vesicles derived from different MSC types and their cargo contents that contribute to corneal tissue regeneration, anti-inflammation, anti-neovascularization, and opacity reduction. Created with BioRender.com under license AU26J1NIGS, assessed on 2 March 2024.

**Figure 6 biomedicines-12-00649-f006:**
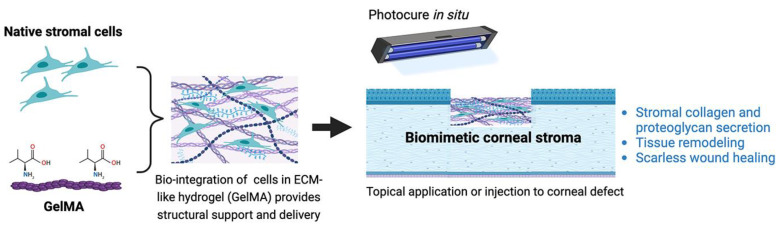
A schematic illustration of bioengineering a biomimetic corneal stroma via integrating native stromal cells (e.g., keratocytes) and ECM-like hydrogel (GelMA). The transplantation to the corneal defect mediates structural restoration of corneal stroma and modulates the stromal matrix environment by proteoglycan secretion to achieve tissue remodeling and scarless wound healing. Created with BioRender.com under license DJ26J1X7F8, assessed on 2 March 2024.

## Data Availability

All data are included in the text.

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
