# Peer review of "Regenerative Therapy for Corneal Scarring Disorders"

_biomedicines, 2024, doi:10.3390/biomedicines12030649_

Round 1

Reviewer 1 Report

Comments and Suggestions for Authors

The review deals with the functions of the cornea, the diseases that can compromise its functions, and the tools available to preserve cornea health and to manage corneal insults. The contents are well organized and the text can be followed easily.

Minor issues

-Page 9, line 383: “which specifically (1) )” The last parenthesis should be omitted.

-Page 10, line 397. Please update the information about the manuscript under review. The authors should ensure that there is not overlapping or literally copy and paste from that manuscript.

-It is unclear whether the authors have permission to reproduce the Figures.

-Section 8.1. Exosomes and extracellular vesicles. Please check that title. Extracellular vesicles include exosomes.

-Could the authors make a comment on platelet-rich plasma?

Author Response

The review deals with the functions of the cornea, the diseases that can compromise its functions, and the tools available to preserve cornea health and to manage corneal insults. The contents are well organized and the text can be followed easily.

Reply - We would like to thank you for the thorough review, helpful comments, and suggestions. Our responses to your comments are outlined below (red highlighted in the revised manuscript).

Minor issues

  1. Page 9, line 383: “which specifically (1) )” The last parenthesis should be omitted.

Reply – Thank you for bringing this to our notice. The typo was removed.

  1. Page 10, line 397. Please update the information about the manuscript under review. The authors should ensure that there is not overlapping or literally copy and paste from that manuscript.

Reply – The manuscript is still under review. There is no plagiarism concerning the data and results described in this study.

  1. It is unclear whether the authors have permission to reproduce the Figures.

Reply – All figures (including new figures) in this manuscript are original. Fig 2, new Fig. 5 and 6 were created with BioRender.com and the publication licenses were included in the legend.

  1. Section 8.1. Exosomes and extracellular vesicles. Please check that title. Extracellular vesicles include exosomes.

Reply – Thank you for bringing this to our notice. The title was revised as “8.1 Extracellular vesicles”.

  1. Could the authors make a comment on platelet-rich plasma?

Reply – Thank you for the suggestion. Platelet-rich plasma (PRP) extracted from blood samples has been shown to consist of high content of growth factors and cell adhesion molecules that are beneficial for tissue healing. In several studies, PRP treatment enhanced corneal wound healing and stimulated epithelial cell proliferation, differentiation, and limbal stemness (Etxebarria 2017 Acta Ophthalmol; Tanidir 2010 Cornea). Collectively, it promotes corneal epithelial regeneration. The clinical use of topical PRP also reversed dormant epithelial ulcers, dry eye syndromes, and ocular surface defects after LASIK surgery (Alio 2012 Curr Pharm Biotechnol; Alio 2017 J Ophthalmol; Rechichi 2020 Am J Ophthalmol Case Rep). However, PRP's effectiveness on stromal wound healing is still controversial. In a rat study, PRP treatment prolonged myofibroblast accumulation by activating TGFβ/SMAD3 signaling (Koulikovska 2015 Curr Eye Res). Hence, PRP seems to be effective in epithelial wound healing and ocular surface regeneration, but its healing ability on stromal wounds and scar inhibition has to be confirmed. The varying contents of growth factors and other healing molecules in PRP samples from different sources could be a possible reason for such controversial effects, even though there are standardized guidelines to prepare PRP with low content of leukocytes and red blood cells (Gomez 2015 Clin Lab). Without data supporting the PRP effects on stromal healing, we prefer not to include it in this review.

Etxebarria et al. Serum from plasma rich in growth factors regenerates rabbit corneas by promoting cell proliferation, migration, differentiation, adhesion and limbal stemness. Acta Ophthalmol. 2017. 95(8):e693-e705. doi: 10.1111/aos.13371.

Tanidir et al. The effect of subconjunctival platelet-rich plasma on corneal epithelial wound healing. Cornea. 2010. 29(6):664-9. doi: 10.1097/ICO.0b013e3181c29633.

Alio et al. The role of "eye platelet rich plasma" (E-PRP) for wound healing in ophthalmology. Curr Pharm Biotechnol. 2012. 13(7):1257-65.doi: 10.2174/138920112800624355.

Alio et al. Autologous Platelet-Rich Plasma Eye Drops for the Treatment of Post-LASIK Chronic Ocular Surface Syndrome. J Ophthalmol 2017. 2017:2457620. doi: 10.1155/2017/2457620.

Rechichi et al. Autologous platelet-rich plasma in the treatment of refractory corneal ulcers: A case report. Am J Ophthalmol Case Rep 2020. 20:100838. doi: 10.1016/j.ajoc.2020.100838.

Koulikovska et al. Platelet-Rich Plasma Prolongs Myofibroblast Accumulation in Corneal Stroma with Incisional Wound. Curr Eye Res 2015. 40(11):1102-10. doi: 10.3109/02713683.2014.978478.

Gomez et al., Standardization of a Protocol for Obtaining Platelet Rich Plasma from blood Donors; a Tool for Tissue Regeneration Procedures. Clin Lab 2015. 61(8):973-80.

Reviewer 2 Report

Comments and Suggestions for Authors

The current review aims to summarize the regenerative therapy for corneal scarring disorders. Although the topic is interesting in its scientific field, there are some issues that require the authors’ attention to improve the quality of this particular manuscript before further consideration for publication in a high-quality journal “Biomedicines”.

Specific comments:

1.      This review article attempted to report the cellular and molecular mechanisms of corneal scarring. The authors should elaborate on the specific roles of different cell types (e.g., corneal cells, myofibroblasts, and inflammatory cells) in scar formation.

2.      In Section 2, the authors conducted an in-depth analysis of the anatomy and physiology of the cornea. However, the information is well-known and should be described in brief.

3.      Given that the current manuscript focused on corneal scar diseases, the authors should give the details about the molecular pathways involved in excessive ECM deposition during corneal tissue remodeling.

4.      Please confirm whether the figures and data in the article are adopted from other paper resources. Please must give appropriate documentation and citation relevant to the original articles.

5.      Although the text is informative, the authors should add more illustrative figures or data to Sections 5-11 in order to facilitate better visualization of complex treatments.

6.      Furthermore, the aforementioned sections described the approaches for treating corneal scar. The authors should also give the limitations and challenges of each part in Section 5-11 to deepen the discussions and input professional viewpoints.

7.      Section 9 described the molecular approach in corneal wound healing. In order to strengthen the manuscript quality, the authors should provide the mechanism diagram of molecular therapy for the treatment of corneal scars.

8.      As stated by the authors, members of TGFβ can either activate or inhibit fibrosis, mechanistically acting through both canonical TGFβ/Smad and non-Smad pathways. However, such an important claim was not supported by appropriate documentation. If possible, please refer to the following example paper (DOI: 10.1002/advs.202302174). In order to balance scientific viewpoint and update article content, the authors are highly recommended to consider the inclusion of this supportive case study in the reference list.

Author Response

The current review aims to summarize the regenerative therapy for corneal scarring disorders. Although the topic is interesting in its scientific field, there are some issues that require the authors’ attention to improve the quality of this particular manuscript before further consideration for publication in a high-quality journal “Biomedicines”.

Reply - We would like to thank the reviewer for the thorough review, comments, and suggestions. Our responses to your comments are outlined below (red highlighted in the revised manuscript).

Specific comments:

  1. This review article attempted to report the cellular and molecular mechanisms of corneal scarring. The authors should elaborate on the specific roles of different cell types (e.g., corneal cells, myofibroblasts, and inflammatory cells) in scar formation.

Reply – Thank you for the suggestion. We added the mechanistic actions of different repair-type stromal cells and the generation of contractile myofibroblasts during wound healing and fibrosis development (on page 4).

  1. In Section 2, the authors conducted an in-depth analysis of the anatomy and physiology of the cornea. However, the information is well-known and should be described in brief.

Reply – Thank you for the suggestion. We have shortened the introduction of the cornea and its anatomy with different cell types (on page 2-3).  

  1. Given that the current manuscript focused on corneal scar diseases, the authors should give the details about the molecular pathways involved in excessive ECM deposition during corneal tissue remodeling.

Reply – We have briefly discussed different fibrosis-associated cell types and signaling pathways, ECM deposition, and scar tissue formation on page 4.

  1. Please confirm whether the figures and data in the article are adopted from other paper resources. Please must give appropriate documentation and citation relevant to the original articles.

Reply – All figures (including new figures) in this manuscript are original. Fig 2, new Fig. 5 and 6 were created with BioRender.com and the publication licenses were included in the legend.

  1. Although the text is informative, the authors should add more illustrative figures or data to Sections 5-11 in order to facilitate better visualization of complex treatments.

Reply – Thank you for the useful suggestion. We added a new Fig. 5 on page 13 to illustrate the sources of extracellular vesicles, cargo properties, and their therapeutic benefits. A new Fig. 6 was added on page 19 to describe the incorporation of cells in GelMA to bioengineer a biomimetic corneal stroma for treating corneal defects. Both figures are original and the publication licenses are stated in the legend.

  1. Furthermore, the aforementioned sections described the approaches for treating corneal scar. The authors should also give the limitations and challenges of each part in Section 5-11 to deepen the discussions and input professional viewpoints.

Reply – Thank you for the suggestion. We added three tables (Table 1, 3, and 4) to describe various methods of cell-based and cell-free strategies, target gene overexpression, and silencing to modulate corneal fibrosis and scarring. In these tables, we discussed the risks and potential side-effects, and the limitations of the approaches.

  1. Section 9 described the molecular approach in corneal wound healing. In order to strengthen the manuscript quality, the authors should provide the mechanism diagram of molecular therapy for the treatment of corneal scars.

Reply – Thank you. In the new Table 3 and 4, we stated the mechanisms of action for each method in modulating corneal fibrosis and scarring.

  1. As stated by the authors, members of TGFβ can either activate or inhibit fibrosis, mechanistically acting through both canonical TGFβ/Smad and non-Smad pathways. However, such an important claim was not supported by appropriate documentation. If possible, please refer to the following example paper (DOI: 10.1002/advs.202302174). In order to balance scientific viewpoint and update article content, the authors are highly recommended to consider the inclusion of this supportive case study in the reference list.

Reply – Thank you for the helpful suggestion. We quoted this study in section 9.7.2 (page 16) to illustrate its impact on achieving a sustained release of pharmacological drugs or growth factors and improving drug bioavailability in injured/inflamed tissues.  

Reviewer 3 Report

Comments and Suggestions for Authors

The article is too focused on the general aspects of corneal physiology and problems rather than focusing on the essential Regenerative therapy for corneal scarring disorders. A table with a summary of therapies that are already in clinical practice and another with what research has been done help with reading. Other aspects that need to be clarified are whether the photos presented in the figures are from the authors' work or obtained from publications.

Author Response

The article is too focused on the general aspects of corneal physiology and problems rather than focusing on the essential Regenerative therapy for corneal scarring disorders. A table with a summary of therapies that are already in clinical practice and another with what research has been done help with reading. Other aspects that need to be clarified are whether the photos presented in the figures are from the authors' work or obtained from publications.

Reply – Thank you for reviewing our manuscript and comments. We added a new Table 2 to summarize the ongoing clinical trials of corneal scar by cell-based and cell-free treatments registered under ClinicalTrial.gov. Information such as target corneal diseases and treatment, current status, and treatment outcomes are included. 

All figures (including new figures) in this revised manuscript are original. Fig 2, new Fig. 5 and 6 were created with BioRender.com and the publication licenses were included in the legend.

Round 2

Reviewer 2 Report

Comments and Suggestions for Authors

The revised version has adequately addressed most of the critiques raised by this reviewer and is now suitable for publication in a high-quality journal “Biomedicines”.

Reviewer 3 Report

Comments and Suggestions for Authors

The authors answered all questions and considered suggestions for improving the manuscript. Thus, the article can be accepted for publication.